# Analysis of Variables That Influence the Walkability of School Environments Based on the Delphi Method

**DOI:** 10.3390/ijerph192114201

**Published:** 2022-10-30

**Authors:** David Cerro-Herrero, Josué Prieto-Prieto, Mikel Vaquero-Solis, Miguel Ángel Tapia-Serrano, Pedro Antonio Sánchez-Miguel

**Affiliations:** 1Department of Didactic of Musical, Plastic and Body Expression, Faculty of Teacher Training, University of Extremadura, 10071 Cáceres, Spain; 2Department of Didactic of Musical, Plastic and Body Expression, School of Education and Tourism, University of Salamanca, 05003 Ávila, Spain; 3Department of Didactic of Musical, Plastic and Body Expression, Faculty of Sport Sciences, University of Extremadura, 10071 Cáceres, Spain

**Keywords:** walkability, school, Delphi method, primary education, children

## Abstract

Walkability is determined the presence or absence of factors such as quality sidewalks, pedestrian crossings, traffic, etc. The ability to walk to the school environment may be one of the variables that promotes active commuting levels. The aim of this study was to examine the walkability of school environments using the Delphi method. This study used the Delphi method to measure the walkability. A total of 18 experts were selected. First, a list of variables was designed by the control group and sent three times to the experts. Later, the items were analyzed qualitatively and quantitatively to test the consensus of the experts. The list of variables that influence walkability showed a good consensus among the experts at the end of the process. This list was formed by 48 items and organized in six factors: traffic and safety (eleven items), signage (eight items), sidewalk (ten items), transport consistency (five items), activity (five items), and finally, urban planning (nine items). The experts agreed on the need to analyze the environments of educational centers and measure the variables that affect walkability. This study has identified the most important barriers. In the future, a measurement instrument should be developed that allows centers to be compared with others in terms of their levels of walkability. Moreover, it might be a resource for more policies to be developed with the aim to promote active commuting to school.

## 1. Introduction

Health benefits of people that commute actively regularly have been shown in research in the context of work, school, and university [1,2]. Despite the significant health implications of active commuting, unfortunately, motorized modes of transport prevail in society [3]. Particularly, in school environments, the frequency of active commuting has declined dramatically in the last few decades; data show that in 1969, 40.7% of students walked or biked to school, and by 2001, the proportion was 12.9% [4]. Spanish adolescent girls in 2007–2008 had lower levels of active commuting to school, especially walking, than their counterparts 6 years before [5].

Active commuting to and from school (ACS) is defined as the use of active modes of transport that imply energy expenditure, such as walking, cycling, skateboarding, or other nonmotorized means [6,7]. Children´s reasons for not commuting actively to school are referred to in previous studies as perceived barriers [8,9,10] and these have become a way for evaluating behavior in children for ACS [11]. It is important to highlight the role of barriers to ACS to implement programs for schools. 

The perception scale of Barriers in Active Transportation to the School (BATACE) [9] validate into Spanish, and adapted from Forman’s [12] studies, classifies barriers from environmental safety planning and psychosocial categories. 

Many studies have examined the perception of psychosocial barriers by students [13,14] and their parents [15,16]; however, few studies have focused on environmental and safety school barriers. 

Examining the environment of schools in terms of the conditions that facilitate active commuting is considered necessary [11]. In this sense, it is essential to present the concept of walkability, which is defined “as the extent to which the characteristics of the built environment and the use of the land may or may not be conducive to walking”, whether to carry out activities of leisure, exercise, recreation, or to access services, travel, or work [17]. 

Walkability is determined by factors such as the presence or absence of quality sidewalks, pedestrian crossings, traffic, obstacles, safety, light, proximity of a variety of basic services, shade, and slope, among others [18]. Studies on walkability have been diverse in recent years, and different measurement methods have been used in school contexts (Table 1).

The analysis of the limitations of the above-mentioned instruments is considered essential to examine the factors that determine the walkability of the school environment. In this purpose, the Delphi method has been selected, as it is considered an effective and systematic procedure [26], which allows for the collection of expert opinions on a particular topic in order to incorporate such judgments in the configuration of an instrument and to achieve a consensus through the convergence of the opinions of geographically dispersed experts [27,28].

In the scope of active commuting and walkability, previous studies have used the Delphi method, such as for validating a questionnaire for parents on the perception of barriers related to active commuting to school [16]. Jittrapirom et al. applied the method for the application of a service to reduce the use of private vehicles based on active mobility, presenting a pilot experience in the Netherlands [29]. Similarly, a scale was designed to measure the walkability of a neighborhood based on a scale with 11 factors (sociodemographic data, soil type, accessibility, connectivity, density, company, services for walkers, comfort, safety, aesthetics, and weather) [30]. Mohamed et al. created an instrument to measure the walkability of urban spaces in Libya [31]. Mandic et al. used a Delphi-like method to propose policy lines to promote active commuting in New Zealand [32]. 

The above-mentioned studies present interesting instruments for inspecting walkability, but they are not adapted to the specificity of school environments and to the barriers that certain elements can pose for moving around urban environments from the view of children and young people. For all these reasons, the objective of the present study is to test the walkability of schools’ environments using the Delphi method.

## 2. Materials and Methods

### 2.1. Delphi Method Application

The previous information collected to pass on to the panel of experts is the result of former questionnaires that measured walkability in school environments (Table 1), including walkability instruments for urban settings, such as in the studies by Mohamed et al. and Ranasinghe et al. [30,31].

The criteria established in previous studies that were developed under the Delphi methodology were followed [33,34]. In addition, the methodological sequence to follow was established, which is made up of three fundamental stages: Preliminary, Exploratory, and Final (Figure 1).

### 2.2. Preliminary Phase

In this phase, the coordinating group was formed, which was in charge of delimiting the study topic and initially formulating the research problem, selecting the pool of potential experts, and securing a collaborative commitment from them.

### 2.3. Selection of Experts

According to the development of the investigation, two groups were defined. On the one hand, the coordinating group was made up of teachers and researchers from the University of Extremadura and the University of Salamanca, who were in charge of agreeing on the results of the second group made up of experts. The group of experts chosen was considered suitable for the development of the research, issuing accurate criteria, and making valid contributions since its members had knowledge based on training and updated experiences. Initially, 53 possible experts were selected, and prior consent to participate was sent to them. Later, 20 experts gave a positive response (Table 2.) As a second step, the procedure called “expert competence coefficient” or “K coefficient” was developed with a view of the self-evaluation by the experts of their competence level in the research topic (Table 3). This procedure is common in studies with the Delphi method [35]. Finally, a total of 18 experts were selected, 10 of whom were considered highly experts because they had skills in walking and in active commuting to schools, and 8 were considered experts for showing skills in one of the two areas. Two experts were excluded from the study because their coefficient of competence (K) was below 0.8 both in walking and in active commuting.

### 2.4. Exploratory Step

In this step, a first list of walkability variables of school environments was designed. Once the first version was designed, the following actions were carried out:The first version was reviewed in a face-to-face meeting by the coordination group, made up of 5 expert researchers in education and physical activity belonging to the Universities of Extremadura and Salamanca. Corrections and adjustments were conducted based on the qualitative criteria that obtained the greatest consensus.The list was sent to the group of experts via email through a process that ensured the anonymity of the experts, and we collected prior acceptance for the participation in the study (Table 4). The experts developed the necessary contributions through a pre-established format that assessed their relevance–adequacy, relevance–importance, and wording–clarity in each item, using a Likert scale from 1 to 5 (1 being the assigned score for the lowest possible value “Not adequate”, 2 for the “Low adequate” value, 3 for the “Adequate” value, 4 for the “Quite adequate” value, and 5 being the assigned value for the highest possible score) which refers to validity. In addition, an open-ended question was asked to collect the qualitative assessments of each expert about every item raised or the introduction of any new item. The maximum period to respond was 10 days.

## 3. Results

### 3.1. First Round

The concordance analyses showed that 23 items (35.94%) fulfilled all the established validity criteria. The relevance–adequacy criteria were fulfilled by 78.13% of the items (n = 50), the relevance–importance criteria were fulfilled by 71.88% (n = 46), and there were 13 items (20.31%) that did not exceed the values in either and therefore were eliminated. A total of 32 items (50%) showed problems regarding writing or clarity (Table 5).

The eliminated items in the first round of the traffic and safety factors were those that referred to the existence of speed radars (item 1.6), existence of areas to stop vehicles (item 1.10), the center having people to accompany the children (item 1.12), and agglomerations of pedestrians around the school (item 1.14). In the signaling factor, items regarding pedestrian areas in the center of street to facilitate crossing (item 2.3) and whether there are direction signs for pedestrians (item 2.10) were eliminated. An item on moving obstacles was removed from the sidewalk factor (item 3.6). In the transportation factor, the item on whether there are free parking areas near the educational center (item 4.6) was eliminated. The activity factor eliminated the items about whether there are people walking with pets (5.3), people greeting each other on the street (item 5.4), and businesses having blinds up (5.11). Finally, the architecture factor eliminated the items that referred to the coherence of some buildings with others (item 6.7) and whether the design of the building is striking (item 6.8).

This section may be divided by subheadings. It should provide a concise and precise description of the experimental results, their interpretation, as well as the experimental conclusions that can be drawn.

After the concordance analysis, a qualitative analysis was carried out based on the narrated discourse of the experts. In this phase, the coordinating group reviewed and adapted the items with the improvements suggested by the experts.

The main contributions made by the experts were in relation to using the term drivers instead of vehicles, using inclusive language, not using ambiguous concepts (much, little), not using overly technical terms of urban planning or architecture (drains), formulating all the items in positive, and finally, the experts suggested the elimination of certain items and the grouping of several others.

Finally, a global analysis of the list was carried out by experts, from which it was concluded that it was necessary to reduce the number of questions. In this regard, the clarity of the approach was considered to be good or excellent by 100% of the experts. Regarding the number of questions, 66.7% considered it regular, and 33.3% considered it good. A total of 60% of the experts considered the list of variables to be suitable. 

The main suggestions provided by the experts globally to the list were:It asks only about the context of the educational center and should be expanded to a larger space since not all schoolchildren live near the school.The last section, instead of architecture, should be titled: “urban morphology and public facilities”.It gives too much importance to road safety and road traffic.It focuses only on the school environment, and that environment may have good walkability conditions, but the movement of students will not only take place in that environment.The size of the environment should be specified: entrance street to the educational center, neighborhood, a radius of one kilometer, etc.Include any variable about slopes.Reduce the number of items.

### 3.2. Second Round

After analyzing the results of the first round, a new version of the list was created. In this new revision, it was specified that it was round 2 and the statistical information obtained from the results of the first round was added. Afterwards, the list was sent by email and the process carried out the first time was repeated. Subsequently sending the expert list and making three email reminders (every 7 days), responses from nine of the selected experts were received.

The concordance analyses at the end of the analysis of the experts’ answers in the second round showed that 41 items (80.39%) met all the established validity criteria. The relevance–adequacy criteria were met by 96.08% of the items (N = 49), the relevance–importance criteria were met by 9.11% (n = 48) and there was only one item (1.96%) that did not exceed the values in either and therefore were eliminated. A total of nine items (17.65%) presented problems regarding writing or clarity (Table 6).

In the second round, the item regarding the accessibility of shops near the educational center was removed from the activity factor (item 5.6), and an item was also removed from those proposed for inclusion that referred to the existence of slopes in the environment of the center (item 7.1).

After the concordance analysis, a qualitative analysis was carried out based on the narrated discourse of the experts. In this phase, the coordinating group reviewed and adapted the items with the improvements suggested by the experts. The main contributions of the experts were “to include the visibility of children as a key aspect of safety”, “to take into account that schools located in pedestrian streets may not meet some items, but despite this, their environment is highly walkable”, “Avoid asking about two concepts on the same items, since one could be fulfilled and the other could not and therefore it is difficult to answer”, and doubts were raised about “whether the facilities for the use of the bicycle favor walking or not”.

At the end of the second round, a global analysis of the list was carried out by the experts, from which it was concluded that it was necessary to reduce the number of variables. Regarding the clarity of the approach, 100% of the experts considered it good or excellent; with respect to the number of questions, 22.2% considered it to be regular and 77.8% good; 10% considered the adequacy of the instrument to the recipients as good or excellent; and finally, 66.7% considered the instructions good and 33.3% considered them excellent.

The main suggestion provided by the experts globally to the questionnaire was that the list was too long.

### 3.3. Final Round

The version obtained at the end of the second round was sent back to the group of experts to qualitatively collect possible suggestions or changes. As a result of this phase, minor grammatical and editorial improvements were made, but the structure of factors and the number of items on the list were not altered (Figure 2).

Some of the most relevant changes proposed were:Establish a Likert scale of five responses with 1, strongly disagreeing; 2, somewhat disagreeing; 3, neutral; 4, somewhat agreeing; and 5, totally agreeing.Eliminate the question format of the items and formulate them in the form of an affirmation to make them more consistent with Likert-type responses.Make references to the time in some items (entrance and exit times to the educational center).Change the name of the architecture factor to urban planning.Write some items in the affirmative and clarify at the end of the questionnaire that their results should be rotated before performing statistical analysis to facilitate the understanding of the questionnaire.Add clarification of the items that must be rotated to obtain the final score (1.10, 3.4, 3.6, 3.9, 6.1, 6.6)

## 4. Discussion

The aim of the present study has been to examine the variables that allow us to evaluate the possibility of schools to assess the walkability of school environments in a simple way through the following factors: traffic and safety, signage, sidewalks, transport, activity, and architecture. The environments of the educational centers were analyzed instead of taking the areas of residence of the students, similar to the study by Macdonald et al. [19].

Previous studies carried out with the aim of evaluating walkability have been subjective, since they were carried out from the participant’s perspective [30,31]. Others are based on the perception of parents as the main factor responsible for the urban mobility of their children [16]. This study was designed to be applied by external experts, thus seeking maximum objectivity in the study environment.

Previous studies on walkability have not evaluated essential factors such as aspects related to safety [19], and others have not been based on scientific methodology [24]. The variable lists designed in this study were elaborated through the Delphi method, starting from an initial version of 60 items that was reduced in the phases of the method to a final version of 48 items. The Delphi method has been widely used for the validation of instruments such as questionnaires or measurement scales, as the opinions obtained by this technique are considered more rigorous and consistent than individual ones [37]. In addition, this technique is a very useful tool when designing and validating new instruments when there is no one that meets the needs of the research to be carried out [38,39].

In this regard, the present study presents a variable list to evaluate the suitability of education centers to carry out interventions that promote active commuting and the benefits obtained from it. This new list may make it possible in the future to create a scale that allows us to measure walkability and compare the results with perceptions of barriers perceived by parents with instruments such as Parental Perception of Barriers towards Active Commuting to School (PABACS) [16] or by students [9], versus with instruments such as the state of the environment observed by external auditors, not influenced by personal variables such as fear for the safety of their children.

However, the present study has limitations, such as the absence of guidelines that mark the consensus among experts [40]. As a strength, this list of variables has been elaborated from the weaknesses of previous ones and from an extensive literary review of studies that value walkability. Likewise, this list provides a scientific tool that gives specific answers to researchers when evaluating the suitability of the centers to carry out an intervention.

## 5. Conclusions

This study concluded that the experts agree on the importance of knowing the variables that influence the walkability of school environments. Similarly, the Delphi method has been shown to provide a high level of consensus among the experts consulted in creating a list of variables that influence walkability.

In line with Betancurth et al. [41], the results of this research are an example of how the Delphi technique provides flexibility in working with experts to guarantee good content validity with a scientific and methodological rigor superior to the methodologies used traditionally, which give greater relevance to psychometric validations without adequate prior content validation, including adaptation.

Future lines of research should use this list to create a scale to measure the walkability level of educational centers through different observers. This instrument, in addition, will allow schools to compare each other, as well as to identify the weak points in terms of walkability and be able to act on them and achieve a much more appropriate environment for active commuting.

## Figures and Tables

**Figure 1 ijerph-19-14201-f001:**
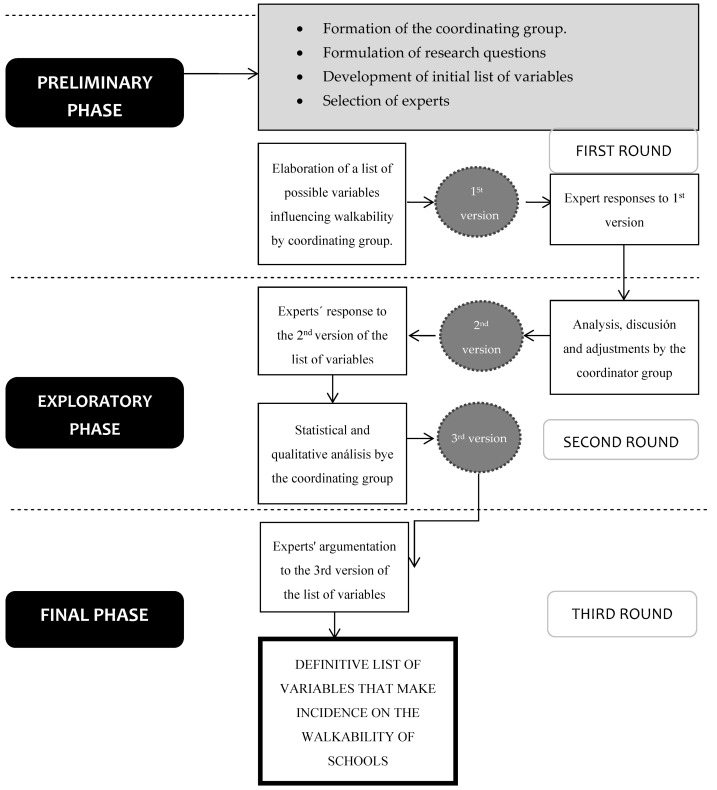
Delphi method description.

**Figure 2 ijerph-19-14201-f002:**
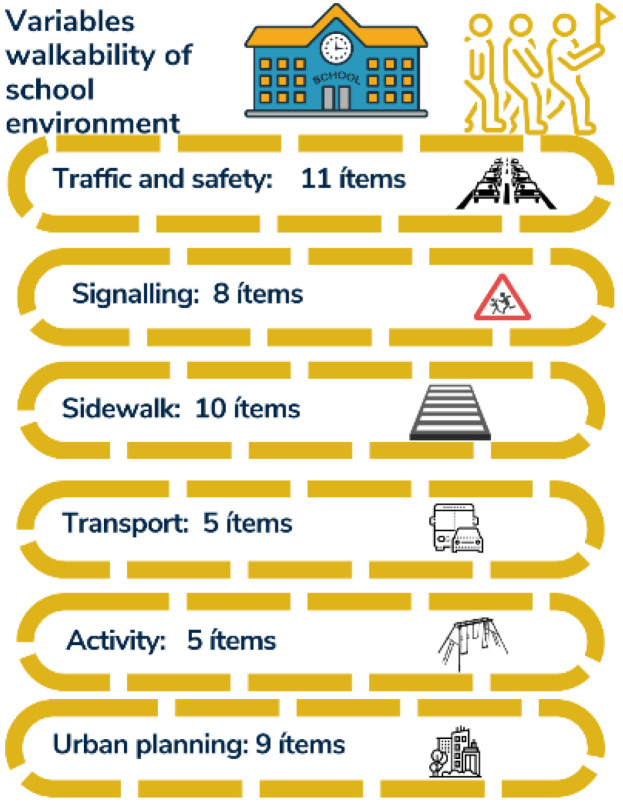
Variables of walkability of school environments.

**Table 1 ijerph-19-14201-t001:** Methods for measuring walkability in school contexts.

Study	State	Measure Method	Limitations (Weak Points)
Macdonald et al. (2019) [19]	Scotland	Walkability score = (2 × intersetions z-scores) + (Housing density z-scores)	Does not consider outcomes such safety, conservation, and other elements (road signs, pedestrian walkways, etc.).
Kim et al. (2016) [20]	United States	Walkability Audit.	Requires several computer applications in addition to interviews with participants.
Moran et al. (2017) [21]	Israel	Walkability index.Include outcomes such as residential density, intersection density, and commercial surface density.	Does not explore aspects specific to educational centers or the state of the infrastructure. Requires the use of geographic information systems.
Vincent et al. (2017) [22]	United States	School walkability scale. Number of intersections/square miles.	Based on numeric data only. Does not consider outcomes such safety, traffic, speed, etc.
Shaaban and Abdur-Rouf (2019) [23]	Qatar	School Audit Tool. Evaluates school environment, road network, parking areas, and active commuting.	The data collection can be made somewhat lengthy by using in each item a description for each value.
Corres and Gonzalez (2018) [24]	Mexico	Audit school walkability. Five dimensions (crosses, velocity, sidewalks, traffic, and safety).	Does not use a rigorous method to design the instrument.In Mexican Spanish language.
Lee et al. (2020) [25]	United States	GIS-based school walkability index.	Requires the use of geographic information systems.

**Table 2 ijerph-19-14201-t002:** Expert characterization and processing results for determining the expert proficiency coefficient.

Expert	Degree	Position	Years of Experience	Walkability			Active Commuting			Assessment
				Kc	Ka	K	Kc	Ka	K	
1	Doctor	University teacher	20	0.80	0.85	0.80	0.80	0.85	0.80	Very high
2	Doctor	University teacher	12	0.80	0.77	0.80	0.90	1.00	0.90	Very high
3	Doctor	Consultant and teacher	34	0.90	0.98	0.90	0.70	0.96	0.80	Very high
4	Doctor	University teacher	11	0.70	0.87	0.80	0.80	0.97	0.90	Very high
5	Doctor	Urban architect	20	0.90	0.98	0.90	0.50	0.63	0.60	High
6	Master’s degree	Project coordinator	5	0.50	0.76	0.60	0.70	0.88	0.80	High
7	Master’s degree	Primary teacher	15	0.90	0.96	0.90	0.90	0.96	0.90	Very high
8	Degree	Secondary teacher	35	0.90	0.74	0.80	1.00	0.84	0.90	Very high
9	Master’s degree	University teacher	20	0.90	0.97	0.90	0.90	0.99	0.90	Very high
10	Doctor	Research fellow	5	0.80	0.89	0.80	0.80	0.89	0.80	Very high
11	Doctor	Investigator	2	0.70	0.65	0.70	0.70	0.78	0.70	Medium
12	Master’s degree	University and primary teacher	17	0.70	0.63	0.70	0.90	0.94	0.90	High
13	Doctorate	University teacher	20	0.90	0.63	0.80	0.90	0.64	0.80	Very high
14	Doctorate	University teacher	2	0.70	0.75	0.70	0.80	0.88	0.80	High
15	Doctorate	University teacher	30	1.00	0.96	0.90	0.80	0.96	0.90	High
16	Master’s degree	Urban architect	11	0.90	0.77	0.80	0.90	0.75	0.80	High
17	Doctorate	Teacher training cycles	21	0.70	0.54	0.60	0.70	0.56	0.60	Medium
18	Master’s degree	Investigator	2	0.70	0.65	0.70	0.80	0.78	0.80	High
19	Doctorate	University teacher	14	0.70	0.79	0.70	0.80	0.78	0.80	High
20	Doctorate	University teacher	6	0.80	0.89	0.80	0.90	1.00	0.90	Very high

Notes: Kc (expert knowledge coefficient), Ka (expert argumentation coefficient), and K (expert competence coefficient). K = ½ (Kc + Ka).

**Table 3 ijerph-19-14201-t003:** Assessment of the sources of reasons to obtain the argumentation coefficient (Ka).

Argumentation Sources	High	Medium	Low
Theoretical analyses developed by you	0.30	0.20	0.10
Experience gained	0.50	0.40	0.20
Studies by national authors you know	0.05	0.04	0.03
Studies by foreign authors you know	0.05	0.04	0.03
Own knowledge about the state of the matter	0.05	0.04	0.03
Your intuition	0.05	0.04	0.03

Note: Source by Cortés and Iglesias [36].

**Table 4 ijerph-19-14201-t004:** Expert evaluation variable list document.

Validation Variable List, First Round
N° Questions/items: 64
Categories or blocks to evaluate Traffic and security (14 items) Signaling (10 items) Sidewalk (11 items) Transportation (6 items) Activity (11 items) Architecture (12 items)
Item evaluation Items were assessed using a 5-point Likert scale with three questions: (1) relevance–adequacy, (2) relevance–importance, and (3) writing–clarity. To be included in the variable list, the criterion adopted to validate the items was that they should be met by the expert evaluations regarding relevance–adequacy and relevance–importance: (1) present a mean greater than 3.75 and a standard deviation less than or equal to 1.5; and (2) present ratings of 4 or 5 in at least 80% of the answers. In each item, an additional box is offered for observations by the experts.
Questionnaire evaluation The clarity of the approach, the number of items, the adequacy of the recipients, and the previous instructions to complete the questionnaire were analyzed. A 4-point Likert scale was used: Bad (M), Regular (R), Good (G), and Excellent (E). An additional box was offered for the proposal of modifications by the experts.

**Table 5 ijerph-19-14201-t005:** Summary of results of round one by factors.

Factor	N° of Initial Items	Relevance–Adequacy			Relevance–Importance			Writing–Clarity			N° of Accepted Items	N° of Accepted Items with Revisions	N° of Reformulated Items	N° of Deleted Items
		M	SD	% 4–5	M	SD	% 4–5	M	SD	% 4–5				
Traffic and safety	14	4.72	0.97	80.48	4.12	1.03	80.48	4.02	1.05	66.65	2.00	7.00	1.00	4.00
Signaling	10	4.67	0.92	83.99	4.18	1.04	77.99	4.07	1.03	69.17	2.00	3.00	3.00	2.00
Sidewalk	11	4.41	0.93	85.46	4.39	0.98	84.05	3.99	1.11	66.92	3.00	7.00	0.00	1.00
Transport	6	4.48	0.99	87.77	4.45	1.07	85.61	4.52	0.81	85.88	5.00	0.00	0.00	1.00
Activity	11	4.31	0.82	83.12	4.23	0.85	80.60	4.35	0.95	78.66	4.00	2.00	2.00	3.00
Architecture	12	4.27	0.75	79.43	4.17	0.80	75.54	4.22	0.96	75.34	7.00	2.00	1.00	2.00

**Table 6 ijerph-19-14201-t006:** Summary of results of the second round by factors.

Factor	N° of Initials Items	Relevance–Adequacy			Relevance–Importance			Writing–Clarity			N° of Accepted Items	N° of Accepted Items with Revisions	N° Of Reformulated Items	N° of Delated Items
		M	SD	% 4–5	M	SD	% 4–5	M	SD	% 4–5				
Traffic and safety	10	4.72	0.49	98.99	4.69	0.49	95.65	4.45	0.88	87.17	7.00	3.00	0.00	0.00
Signaling	8	4.79	0.46	96.21	4.79	0.41	96.21	4.71	0.54	93.56	7.00	1.00	0.00	0.00
Sidewalk	11	4.72	0.52	94.95	4.74	0.51	94.95	4.54	0.78	88.89	8.00	3.00	0.00	0.00
Transport	5	4.76	0.44	99.78	4.78	0.43	99.78	4.89	0.29	99.78	5.00	0.00	0.00	0.00
Activity	6	4.72	0.51	96.47	4.72	0.50	94.47	4.78	0.50	96.47	5.00	0.00	0.00	1.00
Architecture	9	4.77	0.46	99.76	4.75	0.50	97.40	4.87	0.33	100.00	8.00	0.00	1.00	0.00
Traffic and safety	2	4.45	0.96	89.40	4.45	0.96	83.30	4.50	0.85	89.40	1.00	0.00	0.00	1.00

## Data Availability

The data presented in this study are available on request from the corresponding author.

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
