# Peer review of "Analysis of Variables That Influence the Walkability of School Environments Based on the Delphi Method"

_ijerph, 2022, doi:10.3390/ijerph192114201_

Round 1

Reviewer 1 Report

Dear Authors, 

Your article focuses on a theme of current importance, especially at a time when the consequences of the COVID-19 pandemic are still being felt, together with a tendency towards isolation due to the massive use of ICT in school and professional contexts. 

Following on from this thought, the importance of focusing on school children is also highlighted since it is from acquire habits that will condition or enhance their level of health during adulthood.

Apart from the topic, the article is well written, following the structure of a scientific article. However, it will be important to include a paragraph at the end of the introduction presenting the structure of the article. Besides that, the topic of methodology is not properly identified and follows the introduction, with the topics “1.1. Method, design of the investigation, and analysis/ 1.1.1. Delphi method application”. This should appear in an independent and better organised topic. 

In this topic, there is also the need to reorganize the information contained in various topics of the manuscript because there are also several formatting errors that disturb the reading. These should be corrected, particularly regarding the three topics that appear in section "1.2. Exploratory step". These should not be identified with Arabic numerals, but with Roman numerals or letters. Otherwise, it would significantly disrupt the reading of the document. 

Throughout the article, there are also several formal aspects that should be reviewed, namely:

a) the placement of all tables in image format. These should be transformed into tables formatted according to the Journal template

b) the non-identification of the affiliation of the authors

c) in keywords, the word "introduction" appears, which should be deleted 

d) on lines 63 and 172, there is a duplication of the titles of the various tables included in the text

e) revise the text formatting in lines 233 to 235. 

Returning to content issues, the article focuses more on methodological aspects than on finding effective solutions to the problems identified. Although, to some extent, this issue is acknowledged through the considerations made in points 4 and 5 of the article.

Best regards,

The Reviewer

Author Response

David Cerro Herrero

Universidad de Extremadura

Facultad de Formación del Profesorado

Av/ de la Universidad s/n

Cáceres, (España)

+34 699817346

[email protected]

October, 20th 2022

We thank the reviewers for their valuable contributions. All of them have been addressed with the purpose of substantially improving the quality of the manuscript. In particular, all the comments we included are available in the revised manuscript and we have included multiple links throughout the document referring to the modifications made.

We hope that we have correctly responded to all reviewer comments. Please let us know if any other modifications are needed and we will try to modify it as best as possible.

David Cerro Herrero

Professor of the Faculty of Teacher Training, University of Extremadura, Spain

On behalf of all the authors.

Analysis of variables that incidence on walkability of school environment based on Delphi Method

Reviewer#1

Question 1: The placement of all tables in image format. These should be transformed into tables formatted according to the Journal template

Answer 1: Thanks to the reviewer for his or her comment. Following the reviewer's indications, the table of contents has been redesigned as follows:

1.Introduction

  1. Method

2.1 Delphi method application

2.2. Preliminary phase

2.3. Exploratory step

       2.4 Final step

  1. Results

      3.1 First round

      3.2 Second round

      3.3 Final round

  1. Discussion
  2. Conclusions
  3. References

Question 2: the non-identification of the affiliation of the authors.

Answer 2: Thanks to the reviewer for his or her contributions. Tables has been transformed into tables formatted according to the Journal template.

Question 3: on lines 63 and 172, there is a duplication of the titles of the various tables included in the text

Answer 3: Thanks to the reviewer for his or her comment. In keywords, the word "introduction" has been changed to the following paragraph

Question 4: revise the text formatting in lines 233 to 235.

Answer 4: Thanks to the reviewer for his or her contribution. We have revised the text formatting in lines 233 to 235. 

Reviewer 2 Report

Dear authors:

First of all I would like to congratulate you on your great work and to reflect it in this manuscript. Then I would like to suggest some changes to improve your article:

- In the introduction you put the term community, what do you mean. It can be misleading because of its terminology in different countries.

- There is no indexed table in the text, so it cannot be evaluated. I understand that it is a mistake when carrying out the manuscript, but they should resubmit.

- The methodology section is very extensive. I understand that they want to describe the Deplphi study and how they compose the group of experts, but it becomes a bit tedious when it comes to assessing the methodology followed in their study.

- In the results section they should provide more visual information with graphs and images so that the data can be understood without the need for so many words. And they should take into account not to duplicate the information with tables and text.

- The bibliographical references are excessive and should be reduced in this section.

In general the text is of good quality, but in my opinion I think the authors should be more concise and not be so extensive, as many of the things they describe are already understood by an expert audience.

Best regards

Author Response

David Cerro Herrero

Universidad de Extremadura

Facultad de Formación del Profesorado

Av/ de la Universidad s/n

Cáceres, (España)

+34 699817346

[email protected]

October, 20th 2022

We thank the reviewers for their valuable contributions. All of them have been addressed with the purpose of substantially improving the quality of the manuscript. In particular, all the comments we included are available in the revised manuscript and we have included multiple links throughout the document referring to the modifications made.

We hope that we have correctly responded to all reviewer comments. Please let us know if any other modifications are needed and we will try to modify it as best as possible.

David Cerro Herrero

Professor of the Faculty of Teacher Training, University of Extremadura, Spain

On behalf of all the authors.

Analysis of variables that incidence on walkability of school environment based on Delphi Method

Reviewer#2

Question 1: In the introduction you put the term community, what do you mean. It can be misleading because of its terminology in different countries.

Answer 1: Thanks to the reviewer for his or her contributions. We have changed in the introduction community for society.

Question 2: here is no indexed table in the text, so it cannot be evaluated. I understand that it is a mistake when carrying out the manuscript, but they should resubmit.

Answer 2: Thanks to the reviewer for his or her recommendation. Tables and figure have been indexed table in the text.

Question 3: The methodology section is very extensive. I understand that they want to describe the Deplphi study and how they compose the group of experts, but it becomes a bit tedious when it comes to assessing the methodology followed in their study.

Answer 3: Thanks to the reviewer for his or her comment. We agree with the reviewer that the method section is somewhat long, but it is fundamental to understanding the study and its importance. It may also help future researchers to replicate similar studies.

Question 4: In the results section they should provide more visual information with graphs and images so that the data can be understood without the need for so many words. And they should take into account not to duplicate the information with tables and text.

Answer 4: Thanks to the reviewer for his or her contributions. In the results section we have included a new figure to be more and some paragraphs have been deleted because the information already appears in the tables.

Question 5: The bibliographical references are excessive and should be reduced in this section.

Answer 5: Thank you for your comments. In line with your comment, we have reduced the number of references to 41.